# CLEVRER: Collision Events for Video Representation and Reasoning

**Kexin Yi**[*]
Harvard University

**Chuang Gan**[*]
MIT-IBM Watson AI Lab

**Yunzhu Li**
MIT CSAIL

**Pushmeet Kohli**
DeepMind

**Jiajun Wu**
MIT CSAIL

**Antonio Torralba**
MIT CSAIL

**Joshua B. Tenenbaum**
MIT BCS, CBMM, CSAIL

## Abstract

The ability to reason about temporal and causal events from videos lies at the core of human intelligence. Most video reasoning benchmarks, however, focus on pattern recognition from complex visual and language input, instead of on causal structure. We study the complementary problem, exploring the temporal and causal structures behind videos of objects with simple visual appearance. To this end, we introduce the CoLlision Events for Video REpresentation and Reasoning (CLEVRER) dataset, a diagnostic video dataset for systematic evaluation of computational models on a wide range of reasoning tasks. Motivated by the theory of human causal judgment, CLEVRER includes four types of question: descriptive (e.g., 'what color'), explanatory ('what's responsible for'), predictive ('what will happen next'), and counterfactual ('what if'). We evaluate various state-of-the-art models for visual reasoning on our benchmark. While these models thrive on the perception-based task (descriptive), they perform poorly on the causal tasks (explanatory, predictive and counterfactual), suggesting that a principled approach for causal reasoning should incorporate the capability of both perceiving complex visual and language inputs, and understanding the underlying dynamics and causal relations. We also study an oracle model that explicitly combines these components via symbolic representations.

## 1 Introduction

The ability to recognize objects and reason about their behaviors in physical events from videos lies at the core of human cognitive development (Spelke, 2000). Humans, even young infants, group segments into objects based on motion, and use concepts of object permanence, solidity, and continuity to explain what has happened, infer what is about to happen, and imagine what would happen in counterfactual situations. The problem of complex visual reasoning has been widely studied in artificial intelligence and computer vision, driven by the introduction of various datasets on both static images (Antol et al., 2015; Zhu et al., 2016; Hudson & Manning, 2019) and videos (Jang et al., 2017; Tapaswi et al., 2016; Zadeh et al., 2019). However, despite the complexity and variety of the visual context covered by these datasets, the underlying logic, temporal and causal structure behind the reasoning process is less explored.

In this paper, we study the problem of temporal and causal reasoning in videos from a complementary perspective: inspired by a recent visual reasoning dataset, CLEVR (Johnson et al., 2017a), we simplify the problem of visual recognition, but emphasize the complex temporal and causal structure behind the interacting objects. We introduce a video reasoning benchmark for this problem, drawing inspirations from developmental psychology (Gerstenberg et al., 2015; Ullman, 2015). We also evaluate and assess limitations of various current visual reasoning models on the benchmark.

Our benchmark, named CoLlision Events for Video REpresentation and Reasoning (CLEVRER), is a diagnostic video dataset for temporal and causal reasoning under a fully controlled environment. The design of CLEVRER follows two guidelines: first, the posted tasks should focus on logic reasoning in the temporal and causal domain while staying simple and exhibiting minimal biases on visual scenes and language; second, the dataset should be fully controlled and well-annotated in order to host the

---

[*]indicates equal contributions. Project page: http://clevrer.csail.mit.edu/

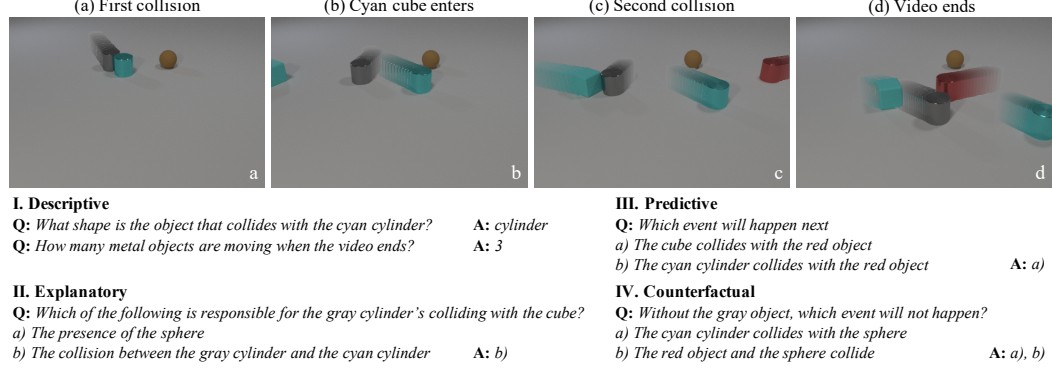

<table>
<tr><td>(a) First collision</td><td>(b) Cyan cube enters</td><td>(c) Second collision</td><td>(d) Video ends</td></tr>
</table>

**I. Descriptive**
**Q:** *What shape is the object that collides with the cyan cylinder?*     **A:** *cylinder*
**Q:** *How many metal objects are moving when the video ends?*     **A:** *3*

**II. Explanatory**
**Q:** *Which of the following is responsible for the gray cylinder's colliding with the cube?*
*a) The presence of the sphere*
*b) The collision between the gray cylinder and the cyan cylinder*     **A:** *b)*

**III. Predictive**
**Q:** *Which event will happen next*
*a) The cube collides with the red object*
*b) The cyan cylinder collides with the red object*     **A:** *a)*

**IV. Counterfactual**
**Q:** *Without the gray object, which event will not happen?*
*a) The cyan cylinder collides with the sphere*
*b) The red object and the sphere collide*     **A:** *a), b)*

Figure 1: Sample video, questions, and answers from our CoLlision Events for Video REpresentation and Reasoning (CLEVRER) dataset. CLEVRER is designed to evaluate whether computational models can understand what is in the video (I, descriptive questions), explain the cause of events (II, explanatory), predict what will happen in the future (III, predictive), and imagine counterfactual scenarios (IV, counterfactual). In the four images (a–d), only for visualization purposes, we apply stroboscopic imaging to reveal object motion. The captions (e.g., 'First collision') are for the readers to better understand the frames, not part of the dataset.

complex reasoning tasks and provide effective diagnostics for models on those tasks. CLEVRER includes 20,000 synthetic videos of colliding objects and more than 300,000 questions and answers (Figure 1). We focus on four specific elements of complex logical reasoning on videos: descriptive (e.g., 'what color'), explanatory ('what's responsible for'), predictive ('what will happen next'), and counterfactual ('what if'). CLEVRER comes with ground-truth motion traces and event histories of each object in the videos. Each question is paired with a functional program representing its underlying logic. As summarized in table 1, CLEVRER complements existing visual reasoning benchmarks on various aspects and introduces several novel tasks.

We also present analysis of various state-of-the-art visual reasoning models on CLEVRER. While these models perform well on descriptive questions, they lack the ability to perform causal reasoning and struggle on the explanatory, predictive, and counterfactual questions. We therefore identify three key elements that are essential to the task: recognition of the objects and events in the videos; modeling the dynamics and causal relations between the objects and events; and understanding of the symbolic logic behind the questions. As a first-step exploration of this principle, we study an oracle model, Neuro-Symbolic Dynamic Reasoning (NS-DR), that explicitly joins these components via a symbolic video representation, and assess its performance and limitations.

## 2 RELATED WORK

Our work can be uniquely positioned in the context of three recent research directions: video understanding, visual question answering, and physical and causal reasoning.

**Video understanding.** With the availability of large-scale video datasets (Caba Heilbron et al., 2015; Kay et al., 2017), joint video and language understanding tasks have received much interest. This includes video captioning (Guadarrama et al., 2013; Venugopalan et al., 2015; Gan et al., 2017), localizing video segments from natural language queries (Gao et al., 2017; Hendricks et al., 2017), and video question answering. In particular, recent papers have explored different approaches to acquire and ground various reasoning tasks to videos. Among those, MovieQA (Tapaswi et al., 2016), TGIF-QA (Jang et al., 2017), TVQA (Lei et al., 2018) are based on real-world videos and human-generated questions. Social-IQ (Zadeh et al., 2019) discusses causal relations in human social interactions based on real videos. COG (Yang et al., 2018) and MarioQA (Mun et al., 2017) use simulated environments to generate synthetic data and controllable reasoning tasks. Compared to them, CLEVRER focuses on the causal relations grounded in object dynamics and physical interactions, and introduces a wide range of tasks including description, explanation, prediction and counterfactuals. CLEVRER also emphasizes compositionality in the visual and logic context.

**Visual question answering.** Many benchmark tasks have been introduced in the domain of visual question answering. The Visual Question Answering (VQA) dataset (Antol et al., 2015) marks an important milestone towards top-down visual reasoning, based on large-scale cloud-sourced real images and human-generated questions. The CLEVR dataset (Johnson et al., 2017a) follows a bottom-up approach by defining the tasks under a controlled close-domain setup of synthetic images and questions with compositional attributes and logic traces. More recently, the GQA dataset (Hudson & Manning, 2019) applies synthetic compositional questions to real images. The VCR dataset (Zellers

| Dataset | Video | Diagnostic Annotations | Temporal Relation | Explanation | Prediction | Counterfactual |
|---|---|---|---|---|---|---|
| VQA (Antol et al., 2015) | ✗ | ✗ | ✗ | ✗ | ✗ | ✗ |
| CLEVR (Johnson et al., 2017a) | ✗ | ✓ | ✗ | ✗ | ✗ | ✗ |
| COG (Yang et al., 2018) | ✗ | ✓ | ✓ | ✗ | ✗ | ✗ |
| VCR (Zellers et al., 2019) | ✗ | ✓ | ✗ | ✓ | ✗ | ✓ |
| GQA (Johnson et al., 2017a) | ✗ | ✓ | ✗ | ✗ | ✗ | ✗ |
| TGIF-QA (Jang et al., 2017) | ✓ | ✗ | ✓ | ✗ | ✗ | ✗ |
| MovieQA (Tapaswi et al., 2016) | ✓ | ✗ | ✓ | ✓ | ✗ | ✗ |
| MarioQA (Mun et al., 2017) | ✓ | ✗ | ✓ | ✓ | ✗ | ✗ |
| TVQA (Lei et al., 2018) | ✓ | ✗ | ✗ | ✓ | ✗ | ✗ |
| Social-IQ (Zadeh et al., 2019) | ✓ | ✗ | ✗ | ✓ | ✗ | ✗ |
| CLEVRER (ours) | ✓ | ✓ | ✓ | ✓ | ✓ | ✓ |

Table 1: Comparison between CLEVRER and other visual reasoning benchmarks on images and videos. CLEVRER is a well-annotated video reasoning dataset created under a controlled environment. It introduces a wide range of reasoning tasks including description, explanation, prediction and counterfactuals

et al., 2019) discusses explanations and hypothesis judgements based on common sense. There have also been numerous visual reasoning models (Hudson & Manning, 2018; Santoro et al., 2017; Hu et al., 2017; Perez et al., 2018; Zhu et al., 2017; Mascharka et al., 2018; Suarez et al., 2018; Cao et al., 2018; Bisk et al., 2018; Misra et al., 2018; Aditya et al., 2018). Here we briefly review a few. The stacked attention networks (SAN) (Yang et al., 2016) introduce a hierarchical attention mechanism for end-to-end VQA models. The MAC network (Hudson & Manning, 2018) combines visual and language attention for compositional visual reasoning. The IEP model (Johnson et al., 2017b) proposes to answer questions via neural program execution. The NS-VQA model (Yi et al., 2018) disentangles perception and logic reasoning by combining an object-based abstract representation of the image with symbolic program execution. In this work we study a complementary problem of causal reasoning and assess the strengths and limitations of these baseline methods.

**Physical and causal reasoning.** Our work is also related to research on learning scene dynamics for physical and causal reasoning (Lerer et al., 2016; Battaglia et al., 2013; Mottaghi et al., 2016; Fragkiadaki et al., 2016; Battaglia et al., 2016; Chang et al., 2017; Agrawal et al., 2016; Finn et al., 2016; Shao et al., 2014; Fire & Zhu, 2016; Pearl, 2009; Ye et al., 2018), either directly from images (Finn et al., 2016; Ebert et al., 2017; Watters et al., 2017; Lerer et al., 2016; Mottaghi et al., 2016; Fragkiadaki et al., 2016), or from a symbolic, abstract representation of the environment (Battaglia et al., 2016; Chang et al., 2017). Concurrent to our work, CoPhy (Baradel et al., 2020) studies physical dynamics prediction in a counterfatual setting. CATER (Girdhar & Ramanan, 2020) introduces a synthetic video dataset for temporal reasoning associated with compositional actions. CLEVRER complements these works by incorporating dynamics modeling with compositional causal reasoning, and grounding the reasoning tasks to the language domain.

## 3 THE CLEVRER DATASET

The CLEVRER dataset studies temporal and causal reasoning on videos. It is carefully designed in a fully-controlled synthetic environment, enabling complex reasoning tasks, providing effective diagnostics for models while simplifying video recognition and language understanding. The videos describe motion and collisions of objects on a flat tabletop (as shown in Figure 1) simulated by a physics engine, and are associated with the ground-truth motion traces and histories of all objects and events. Each video comes with four types of questions generated by machine, including descriptive ('what color', 'how many'), explanatory ('What is responsible for'), predictive ('What will happen next'), and counterfactual ('what if'). Each question is paired with a functional program.

### 3.1 VIDEOS

CLEVRER includes 10,000 videos for training, 5,000 for validation, and 5,000 for testing. All videos last for 5 seconds. The videos are generated by a physics engine that simulates object motion plus a graphs engine that renders the frames. Extra examples from the dataset can be found in supplementary material C.

**Objects and events.** Objects in CLEVRER videos adopt similar compositional intrinsic attributes as in CLEVR (Johnson et al., 2017a), including three *shapes* (cube, sphere, and cylinder), two *materials* (metal and rubber), and eight *colors* (gray, red, blue, green, brown, cyan, purple, and

*How many metal objects are moving when the video ends?*

*Without the gray object, which event will happen?*

*The cube collides with the red object.*

Figure 2: Sample questions and programs from CLEVRER. Left: Descriptive question. Middle and right: multiple-choice question and choice. Each choice can pair with the question to form a joint logic trace.

yellow). All objects have the same size so no vertical bouncing occurs during collision. In each video, we prohibit identical objects, such that each combination of the three attributes uniquely identifies one object. Under this constraint, all intrinsic attributes for each object are sampled randomly. We further introduce three types of events: *enter*, *exit* and *collision*, each of which contains a fixed number of object participants: 2 for collision and 1 for enter and exit. The objects and events form an abstract representation of the video. These ground-truth annotations, together with the object motion traces, enable model diagnostics, one of the key advantages offered by a fully controlled environment.

**Causal structure.**  Objects and events in CLEVRER videos exhibit rich causal structures. An event can be either caused by an object if the event is the first one participated by the object, or another event if the cause event happens right before the outcome event on the same object. For example, if a sphere collides with a cube and then a cylinder, then the first collision and the cube jointly "cause" the second collision. The object motion traces with complex causal structures are generated by the following recursive process. We start with one randomly initialized moving object, and then add another object whose initial position and velocity are set such that it will collide with the first object. The same process is then repeated to add more objects and collisions to the scene. All collision pairs and motion trajectories are randomly chosen. We discard simulations with repetitive collisions between the same pair of objects.

**Video generation.**  CLEVRER videos are generated from the simulated motion traces, including each object's position and pose at each time step. We use the Bullet (Coumans, 2010) physics engine for motion simulation. Each simulation lasts for seven seconds. The motion traces are first down-sampled to fit the frame rate of the output video (25 frames per second). Then the motion of the first five seconds are sent to Blender (Blender Online Community, 2016) to render realistic video frames of object motion and collision. The remaining two seconds are held-out for predictive tasks. We further note CLEVRER adopts the same software and parameters for rendering as CLEVR.

## 3.2 QUESTIONS

We pair each video with machine-generated questions for descriptive, explanatory, predictive, and counterfactual reasoning. Sample questions of the four types can be found in Figure 1. Each question is paired with a functional program executable on the video's dynamical scene. Unlike CLEVR (Johnson et al., 2017a), our questions focus on the temporal and causal aspects of the objects and events in the videos. We exclude all questions on static object properties, which can be answered by looking at a single frame. CLEVRER consists of 219,918 descriptive questions, 33,811 explanatory questions, 14,298 predictive questions and 37,253 counterfactual questions. Detailed distribution and split of the questions can be found in Figure 3 and supplementary material A.

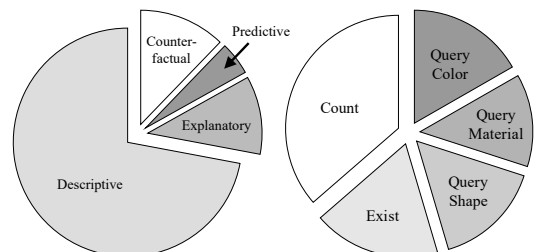

Figure 3: Distribution of CLEVRER question types. Left: distribution of four main questions types. Right: distribution of descriptive sub-types.

**Descriptive.**  Descriptive questions evaluate a model's capability to understand and reason about a video's dynamical content and temporal relation. The reasoning tasks are grounded to the compositional space of both object and event properties, including intrinsic attributes (color, material, shape), motion, collision, and temporal order. All descriptive questions are 'open-ended' and can be answered by a single word. Descriptive questions contain multiple sub-types including *count*, *exist*, *query color*, *query material*, and *query shape*. Distribution of the sub-types is shown in Figure 3. We evenly sample the answers within each sub-type to reduce answer bias.

**Explanatory.**  Explanatory questions query the causal structure of a video by asking whether an object or event is responsible for another event. Event $A$ is responsible for event $B$ if $A$ is among

$B$'s ancestors in the causal graph. Similarly, object $O$ is responsible for event $A$ if $O$ participates in $A$ or any other event responsible for $A$. Explanatory questions are multiple choice questions with at most four options, each representing an event or object in the video. Models need to select all options that match the question's description. There can be multiple correct options for each question. We sample the options to balance the number of correct and wrong ones, and minimize text-only biases.

**Predictive.** Predictive questions test a model's capability of predicting possible occurrences of future events after the video ends. Similar to explanatory questions, predictive questions are multiple-choice, whose options represent candidate events that will or will not happen. Because post-video events are sparse, we provide two options for each predictive question to reduce bias.

**Counterfactual.** Counterfactual questions query the outcome of the video under certain hypothetical conditions (e.g. removing one of the objects). Models need to select the events that would or would not happen under the designated condition. There are at most four options for each question. The numbers of correct and incorrect options are balanced. Both predictive and counterfactual questions require knowledge of object dynamics underlying the videos and the ability to imagine and reason about unobserved events.

**Program representation.** In CLEVRER, each question is represented by a tree-structured functional program, as shown in Figure 2. A program begins with a list of objects or events from the video. The list is then passed through a sequence of filter modules, which select entries from the list and join the tree branches to output a set of target objects and events. Finally, an output module is called to query a designated property of the target outputs. For multiple choice questions, each question and option correspond to separate programs, which can be jointly executed to output a yes/no token that indicates if the choice is correct for the question. A list of all program modules can be found in the supplementary material B.

**Question generation.** Questions in CLEVRER are generated by a multi-step procedure. For each question type, a pre-defined logic template is chosen. The logic template can be further populated by attributes that are associated with the context of the video (i.e. the color, material, shape that identifies the object to be queried) and then turned into natural language. We first generate an exhaustive list of all possible questions for each video. To avoid language bias, we sample from that list to maintain a balanced answer distribution for each question type and minimize the correlation between the questions and answers over the entire dataset.

## 4 BASELINE EVALUATION

In this section, we evaluate and analyse the performances of a wide range of baseline models for video reasoning on CLEVRER. For descriptive questions, the models treat each question as a multi-class classification problem over all possible answers. For multiple choice questions, each question-choice pair is treated as a binary classification problem indicating the correctness of the choice.

### 4.1 MODEL DETAILS

The baseline models we evaluate fall into three families: language-only models, models for video question answering, and models for compositional visual reasoning.

**Language-only models.** This model family includes weak baselines that only relies on question input to assess language biases in CLEVRER. **Q-type (random)** uniformly samples an answer from the answer space or randomly select each choice for multiple-choice questions. **Q-type (frequent)** chooses the most frequent answer in the training set for each question type. **LSTM** uses a pretrained word embedding trained on the Google News corpus (Mikolov et al., 2013) to encode the input question and processes the sequence with a LSTM (Hochreiter & Schmidhuber, 1997). A MLP is then applied to the final hidden state to predict a distribution over the answers.

**Video question answering.** We also evaluate the following models that relies on both video and language inputs. **CNN+MLP** extracts features from the input video via a convolutional neural network (CNN) and encodes the question by taking the average of the pretrained word embeddings (Mikolov et al., 2013). The video and language features are then jointly sent to a MLP for answer prediction. **CNN+LSTM** relies on the same architecture for video feature extraction but uses the final state of a LSTM for answer prediction. TVQA (Lei et al., 2018) introduces a multi-stream end-to-end neural model that sets the state of the art for video question answering. We apply attribute-aware object-centric features acquired by a video frame parser (**TVQA+**). We also include a recent model that incorporates heterogeneous memory with multimodal attention work (**Memory**) (Fan et al., 2019) that achieves superior performance on several datasets.

| Methods | Descriptive | Explanatory | | Predictive | | Counterfactual | |
|---|---|---|---|---|---|---|---|
| | | per opt. | per ques. | per opt. | per ques. | per opt. | per ques. |
| Q-type (random) | 29.2 | 50.1 | 8.1 | 50.7 | 25.5 | 50.1 | 10.3 |
| Q-type (frequent) | 33.0 | 50.2 | 16.5 | 50.0 | 0.0 | 50.2 | 1.0 |
| LSTM | 34.7 | 59.7 | 13.6 | 50.6 | 23.2 | 53.8 | 3.1 |
| CNN+MLP | 48.4 | 54.9 | 18.3 | 50.5 | 13.2 | 55.2 | 9.0 |
| CNN+LSTM | 51.8 | 62.0 | 17.5 | 57.9 | 31.6 | 61.2 | 14.7 |
| TVQA+ | 72.0 | 63.3 | 23.7 | 70.3 | 48.9 | 53.9 | 4.1 |
| Memory | 54.7 | 53.7 | 13.9 | 50.0 | 33.1 | 54.2 | 7.0 |
| IEP (V) | 52.8 | 52.6 | 14.5 | 50.0 | 9.7 | 53.4 | 3.8 |
| TbD-net (V) | 79.5 | 61.6 | 3.8 | 50.3 | 6.5 | 56.1 | 4.4 |
| MAC (V) | 85.6 | 59.5 | 12.5 | 51.0 | 16.5 | 54.6 | 13.7 |
| MAC (V+) | 86.4 | 70.5 | 22.3 | 59.7 | 42.9 | 63.5 | 25.1 |

Table 2: Question-answering accuracy of visual reasoning baselines on CLEVRER. All models are trained on the full training set. The IEP (V) model and TbD-net (V) use 1000 programs to train the program generator.

**Compositional visual reasoning.** The CLEVR dataset (Johnson et al., 2017a) opened up a new direction of compositional visual reasoning, which emphasizes complexity and compositionality in the logic and visual context. We modify several best-performing models and apply them to our video benchmark. The IEP model (Johnson et al., 2017b) applies neural program execution for visual reasoning on images. We apply the same approach to our program-based video reasoning task (**IEP (V)**) by substituting the program primitives by the ones from CLEVRER, and applying the execution modules on the video features extracted by a convolutional LSTM (Shi et al., 2015). TbD-net (Mascharka et al., 2018) follows a similar approach by parsing the input question into a program, which is then assembled into a neural network that acts on the *attention map* over the image features. The final attended image feature is then sent to an output layer for classification. We adopt the same approach through spatial-temporal attention over the video feature space (**TbD-net (V)**). MAC (Hudson & Manning, 2018) incorporates a joint attention mechanism on both the image feature map and the question, which leads to strong performance on CLEVR without program supervision. We modify the model by applying a temporal attention unit across the video frames to generate a latent encoding for the video (**MAC (V)**). The video feature is then input to the MAC network to output an answer distribution. We study an augmented approach: we construct object-aware video features by adding the segmentation masks of all objects in the frames and labeling them by the values of their intrinsic attributes (**MAC (V+)**).

**Implementation details.** We use a pre-trained ResNet-50 (He et al., 2016) to extract features from the video frames. We use the 2,048-dimensional pool5 layer output for CNN-based methods, and the $14 \times 14$ feature maps for MAC, IEP and TbD-net. The program generators of IEP and TbD-net are trained on 1000 programs. We uniformly sample 25 frames for each video as input. Object segmentation masks and attributes are obtained by a video parser consisted of an object detector and an attribute network.

## 4.2 RESULTS

We summarize the performances of all baseline models in Table 2. The fact that these models achieve different performances over the wide spectrum of tasks suggest that CLEVRER offers powerful assessment to the models' strength and limitations on various domains. All models are trained on the training set until convergence, tuned on the validation set and evaluated on the test set.

**Evaluation metrics.** For descriptive questions, we calculate the accuracy by comparing the predicted answer token to the ground-truth. For multiple choice questions, we adopt two metrics: per-option accuracy measures the model's overall correctness on single options across all questions; per-question accuracy measures the correctness of the full question, requiring all choices to be selected correctly.

**Descriptive reasoning.** Descriptive questions query the content of the video from various aspects. In order to do well on this question type, a model needs to both accurately recognize the objects and events that happen in the video, as well as understanding the compositional logic pattern behind the questions. In other words, descriptive questions require strong perception and logic operations on both visual and language signals. As shown in Table 2, the LSTM baseline that relies only on question input performs poorly on the descriptive questions, only outperforming the random baselines

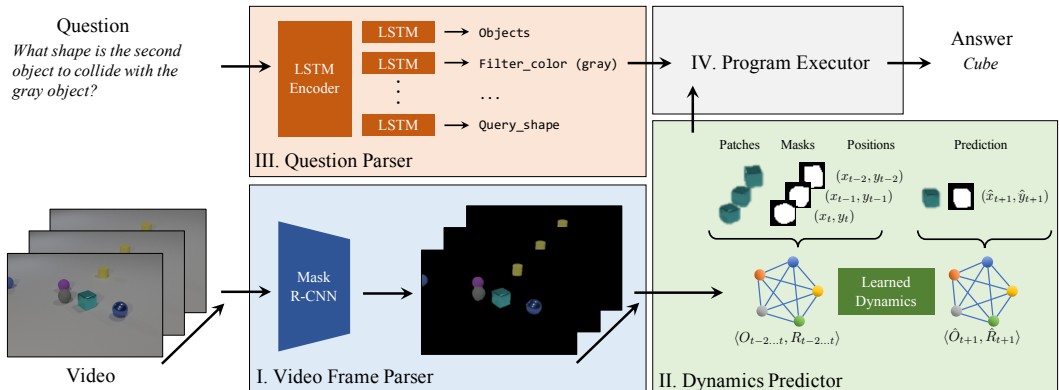

Figure 4: Our model includes four components: a video frame parser that generates an object-based representation of the video frames; a question parser that turns a question into a functional program; a dynamics predictor that extracts and predicts the dynamic scene of the video; and a symbolic program executor that runs the program on the dynamic scene to obtain an answer.

by a small margin. This suggests that CLEVRER has very small bias on the questions. Video QA models, including the state of the art model TVQA+ (Lei et al., 2018) achieve better performances. But because of their limited capability of handling the compositionality in the question logic and visual context, these models are still unable to thrive on the task. In contrast, models designed for compositional reasoning, including TbD-net that operates on neural program execution and MAC that introduces a joint attention mechanism, are able to achieve more competitive performances.

**Causal reasoning.**    Results on the descriptive questions demonstrate the power of models that combine visual and language perception with compositional logic operations. However, the causal reasoning tasks (explanatory, predictive, counterfactual) require further understanding beyond perception. Our evaluation results (Table 2) show poor performance of most baseline models on these questions. The compositional reasoning models that performs well on the descriptive questions (MAC (V) and TbD-net (V)) only achieve marginal improvements over the random and language-only baselines on the causal tasks. However, we do notice a reasonable gain in performance on models that inputs object-aware representations: TVQA+ achieves high accuracy on the predictive questions, and MAC (V+) improves upon MAC (V) on all tasks.

We identify the following messages suggested by our evaluation results. First, object-centric representations are essential for the causal reasoning tasks. This is supported by the large improvement on MAC (V+) over MAC (V) after using features that are aware of object instances and attributes, as well as the strong performance of TVQA+ on the predictive questions. Second, all baseline models lack a component to explicitly model the dynamics of the objects and the causal relations between the collision events. As a result, they struggle on the tasks involving unobserved scenes and in particular performs poorly on the counterfactual tasks. The combination of object-centric representation and dynamics modeling therefore suggests a promising direction for approaching the causal tasks.

## 5    NEURO-SYMBOLIC DYNAMIC REASONING

Baseline evaluations on CLEVRER have revealed two key elements that are essential to causal reasoning: an object-centric video representation that is aware of the temporal and causal relations between the objects and events; and a dynamics model able to predict the object dynamics under unobserved or counterfactual scenarios. However, unifying these elements with video and language understanding posts the following challenges: first, all the disjoint model components should operate on a common set of representations of the video, question, dynamics and causal relations; second, the representation should be aware of the compositional relations between the objects and events. We draw inspirations from Yi et al. (2018) and study an oracle model that operates on a *symbolic* representation to join video perception, language understanding with dynamics modeling. Our model Neuro-Symbolic Dynamic Reasoning (NS-DR) combines neural nets for pattern recognition and dynamics prediction, and symbolic logic for causal reasoning. As shown in Figure 4, NS-DR consists of a video frame parser (Figure 4-I), a neural dynamics predictor (Figure 4-II), a question parser (Figure 4-III), and a program executor (Figure 4-IV). We present details of the model components below and in supplementary material B.

**Video frame parser.** The video frame parser serves as a perception module that is disentangled from other components of the model, from which we obtain an object-centric representation of the video frames. The parser is a Mask R-CNN (He et al., 2017) that performs object detection and scene de-rendering on each video frame (Wu et al., 2017). We use a ResNet-50 FPN (Lin et al., 2017; He et al., 2016) as the backbone. For each input video frame, the network outputs the object mask proposals, a label that corresponds to the intrinsic attributes (color, material, shape) and a score representing the confidence level of the proposal used for filtering. Please refer to He et al. (He et al., 2017) for more details. The video parser is trained on 4000 video frames randomly sampled from the training set with object masks and attribute annotations.

**Neural dynamics predictor.** We apply the Propagation Network (PropNet) (Li et al., 2019) for dynamics modeling, which is a learnable physics engine that, extending from the interaction networks (Battaglia et al., 2016), performs object- and relation-centric updates on a dynamic scene. The model inputs the object proposals from the video parser, and learns the dynamics of the objects across the frames for predicting motion traces and collision events. PropNet represents the dynamical system as a directed graph, $G = \langle O, R \rangle$, where the vertices $O = \{o_i\}$ represent objects and edges $R = \{r_k\}$ represent relations. Each object (vertex) $o_i$ and relation (edge) $r_k$ can be further written as $o_i = \langle s_i, a_i^o \rangle$, $r_k = \langle u_k, v_k, a_k^r \rangle$, where $s_i$ is the state of object $i$; $a_i^o$ denotes its intrinsic attributes; $u_k, v_k$ are integers denoting the index of the receiver and sender vertices joined by edge $r_k$; $a_k^r$ represents the state of edge $r_k$, indicating whether there is collision between the two objects. In our case, $s_i$ is a concatenation of tuple $\langle c_i, m_i, p_i \rangle$ over a small history window to encode motion history, where $c_i$ and $m_i$ are the corresponding image and mask patches cropped at $p_i$, which is the $x$-$y$ position of the mask in the original image (please see the cyan metal cube in Figure 4 for an example). PropNet handles the instantaneous propagation of effects via multi-step message passing. Please refer to supplementary material B for more details.

**Question parser.** We use an attention-based seq2seq model (Bahdanau et al., 2015) to parse the input questions into their corresponding functional programs. The model consists of a bidirectional LSTM encoder plus an LSTM decoder with attention, similar to the question parser in (Yi et al., 2018). For multiple choice questions, we use two networks to parse the questions and choices separately. Given an input word sequence $(x_1, x_2, ...x_I)$, the encoder first generates a bi-directional latent encoding at each step:

$$e_i^f, h_i^f = \text{LSTM}(\Phi_I(x_i), h_{i-1}^f), \tag{1}$$

$$e_i^b, h_i^b = \text{LSTM}(\Phi_I(x_i), h_{i+1}^b), \tag{2}$$

$$e_i = [e_i^f, e_i^b]. \tag{3}$$

The decoder then generates a sequence of program tokens $(y_1, y_2, ..., y_J)$ from the latent encodings using attention:

$$v_j = \text{LSTM}(\Phi_O(y_{j-1})), \tag{4}$$

$$\alpha_{ji} \propto \exp(v_j^T e_i), \qquad c_j = \sum_i \alpha_{ji} e_i, \tag{5}$$

$$\hat{y}_j \sim \text{softmax}(W \cdot [q_j, c_j]). \tag{6}$$

At training time, the input program tokens $\{y_j\}$ are used for generating the predicted labels $y_{j-1} \rightarrow \hat{y}_j$ at each step $j = 1, 2, 3, ..., J$. The generated label $\hat{y}_j$ is then compared with $y_j$ to compute a loss for training. At test time, the decoder rolls out by feeding the sampled prediction at the previous step to the input of the current step $\hat{y}_j = y_j$. The roll-out stops when the end token appears or when the sequence reaches certain maximal length. For both the encoder and decoder LSTM, we use the same parameter setup of two hidden layers with 256 units, and a 300-dimensional word embedding vector.

**Program executor.** The program executor explicitly executes the program on the motion and event traces extracted by the dynamics predictor and output an answer to the question. It consists of a collection of functional modules implemented in Python. Given an input program, the executor first assembles the modules and then iterate through the program tree. The output of the final module is the answer to the target question. There are three types of program modules: input module, filter module, and output module. The input modules are the entry points of the program trees and can directly query the output of the dynamics predictor. For example, the `Events` and `Objects` modules that commonly appear in the programs of descriptive questions indicate inputting all the observed events and objects from a video. The filter modules perform logic operations on the input objects/events

| Methods | Descriptive | Explanatory | | Predictive | | Counterfactual | |
|---------|-------------|-------------|-----------|------------|-----------|----------------|-----------|
| | | per opt. | per ques. | per opt. | per ques. | per opt. | per ques. |
| NS-DR | 88.1 | 87.6 | 79.6 | 82.9 | 68.7 | 74.1 | 42.2 |
| NS-DR (NE) | 85.8 | 85.9 | 74.3 | 75.4 | 54.1 | 76.1 | 42.0 |

Table 3: Quantitative results of NS-DR on CLEVRER. We evaluate our model on all four question types. We also study a variation of our model NS-DR (NE) with "no events" but only motion traces from the dynamics predictor. Our question parser is trained with 1000 programs.

based on a designated intrinsic attribute, motion state, temporal order, or causal relation. The output modules return the answer label. For open-ended descriptive questions, the executor will return a token from the answer space. For multiple choice questions, the executor will first execute the choice program and then send the result to the question program to output a yes/no token. A comprehensive list of all modules in the NS-DR program executor is presented in supplementary material B.

**Results.** We summarize the performance of NS-DR on CLEVRER in Table 3. On descriptive questions, our model achieves an 88.1% accuracy when the question parser is trained under 1,000 programs, outperforming other baseline methods. On explanatory, predictive, and counterfactual questions, our model achieves a more significant gain. We also study a variation of our model, NS-DR (No Events/NE), which uses the dynamics predictor to generate only the predictive and counterfactual motion traces, and collision events are identified by the velocity change of the colliding objects. NS-DR (NE) performs comparably to the full model, showing that our disentangled system is adaptable to alternative methods for dynamics modeling. We also conduct ablation study on the number of programs for training the question parser. As shown in Figure 5, NS-DR reaches full capability at 1,000 programs for all question types.

We highlight the following contributions of the model. First, NS-DR incorporates a dynamics planner into the visual reasoning task, which directly enables predictions of unobserved motion and events, and enables the model for the predictive and counterfactual tasks. This suggests that dynamics planning has great potential for language-grounded visual reasoning tasks and NS-DR takes a preliminary step towards this direction. Second, symbolic representation provides a powerful common ground for vision, language, dynamics and causality. By design, it empowers the model to explicitly capture the compositionality behind the video's causal structure and the question logic.

We further discuss limitations of NS-DR and suggest possible directions for future research. First, training of the video and question parser relies on extra supervisions such as object masks, attributes, and question programs. Even though the amount of data required for training is minimal compared to end-to-end approaches (i.e. thousands of annotated frames and programs), these data is hard to acquire in real-world applications. This constraint could be relaxed by applying unsupervised/weakly-supervised methods for scene decomposition and concept discovery (Burgess et al., 2019; Mao et al., 2019). Second, our model performance decreases on tasks that require long-term dynamics prediction such as the

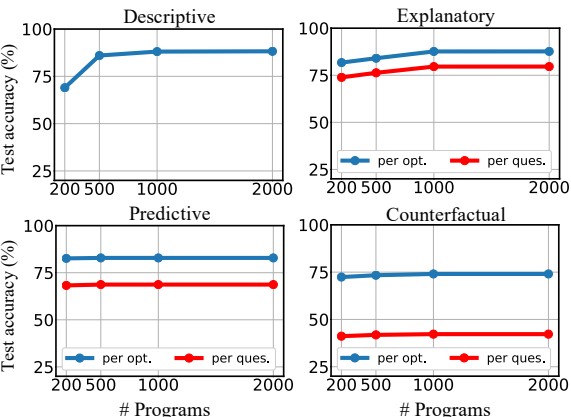

Figure 5: Performance of our model under different number of programs used for training the question parser.

counterfactual questions. This suggests that we need a better dynamics model capable of generating more stable and accurate trajectories. CLEVRER provides a benchmark for assessing the predictive power of such dynamics models.

## 6 CONCLUSION

We present a systematic study of temporal and causal reasoning in videos. This profound and challenging problem deeply rooted to the fundamentals of human intelligence has just begun to be studied with 'modern' AI tools. We introduce a set of benchmark tasks to better facilitate the research in this area. We also believe video understanding and reasoning should go beyond passive knowledge extraction, and focus on building an internal understanding of the dynamics and causal relations,

which is essential for practical applications such as dynamic robot manipulation under complex causal conditions. Our newly introduced CLEVRER dataset and the NS-DR model are preliminary steps toward this direction. We hope that with recent advances in graph networks, visual predictive models, and neuro-symbolic algorithms, the deep learning community can now revisit this classic problem in more realistic setups in the future, capturing true intelligence beyond pattern recognition.

**Acknowledgment.** We thank Jiayuan Mao for helpful discussions and suggestions. This work is in part supported by ONR MURI N00014-16-1-2007, the Center for Brain, Minds, and Machines (CBMM, funded by NSF STC award CCF-1231216), and IBM Research.

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

# SUPPLEMENTARY MATERIALS

## A  DESCRIPTIVE QUESTION SUB-TYPES AND STATISTICS

Descriptive questions in CLEVRER consists of five sub-types. When generating the questions, we balance the frequency of the answers within each question sub-type to reduce bias. The sub-types and their answer space distribution are shown in Figure

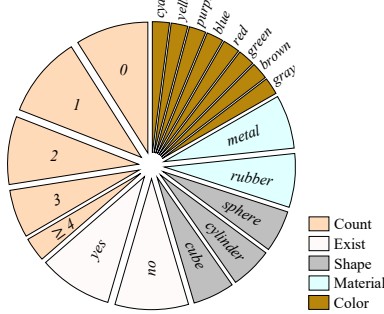

| Sub-type | Answers |
|----------|---------|
| Count | *0, 1, 2, 3, 4, 5* |
| Exist | *yes, no* |
| Query_material | *rubber, metal* |
| Query_shape | *sphere, cube, cylinder* |
| Query_color | *gray, brown, green, red, blue, purple, yellow, cyan* |

Figure 6: Descriptive question sub-type and answer space statistics.

## B  NS-DR MODEL DETAILS AND TRAINING PARADIGM

Here we present further details of the NS-DR model and the training paradigm. A running example of the model can be found in Figure 7.

**Video frame parser.**  Our frame parser is trained on 4,000 frames randomly selected from the training videos plus ground-truth masks and attribute annotations of each object in the frames. We train the model for 30,000 iterations with stochastic gradient decent, using a batch size of 6 and learning rate 0.001. At test time, we keep the object proposals with a confidence score of $> 0.9$ and use those to form the object-level abstract representation of the video.

**Neural dynamics predictor.**  The object encoder $f_O^{\text{enc}}$ and the relation encoder $f_R^{\text{enc}}$ in PropNet are instantiated as convolutional neural networks and output a $D_{\text{enc}}$-dim vector as the representation. We add skip connections between the object encoder $f_O^{\text{enc}}$ and the predictor $f_O^{\text{pred}}$ for each object to generate higher quality images. We also include a relation predictor $f_R^{\text{pred}}$ to determine whether two objects will collide or not in the next time step.

At time $t$, we first encodes the objects and relations $c_{i,t}^o = f_O^{\text{enc}}(o_{i,t}), c_{k,t}^r = f_R^{\text{enc}}(o_{u_k,t}, o_{v_k,t}, a_k^r)$, where $o_{i,t}$ denotes object $i$ at time $t$. We then denotes the propagating effect from relation $k$ at propagation step $l$ as $e_{k,t}^l$, and the effect from object $i$ as $h_{i,t}^l$. For step $1 \leq l \leq L$, the propagation procedure can be described as

Step 0:
$$h_{i,t}^0 = \mathbf{0}, \quad i = 1 \ldots |O|, \tag{7}$$

Step $l = 1, \ldots, L$:
$$e_{k,t}^l = f_R(c_{k,t}^r, h_{u_k,t}^{l-1}, h_{v_k,t}^{l-1}), \quad k = 1 \ldots |R|,$$
$$h_{i,t}^l = f_O(c_{i,t}^o, \sum_{k \in \mathcal{N}_i} e_{k,t}^l, h_{i,t}^{l-1}), \quad i = 1 \ldots |O|, \tag{8}$$

Output:
$$\hat{r}_{k,t+1} = f_R^{\text{pred}}(c_{k,t}^r, e_{k,t}^L), \quad k = 1 \ldots |R|,$$
$$\hat{o}_{i,t+1} = f_O^{\text{pred}}(c_{i,t}^o, h_{i,t}^L), \quad i = 1 \ldots |O| \tag{9}$$

where $f_O$ denotes the object propagator, $f_R$ denotes the relation propagator, and $\mathcal{N}_i$ denotes the relation set where object $i$ is the receiver. The neural dynamics predictor is trained by minimizing the $\mathcal{L}_2$ distance between the predicted $\hat{r}_{k,t+1}, \hat{o}_{i,t+1}$ and the real future observation $r_{k,t+1}, o_{i,t+1}$ using stochastic gradient descent.

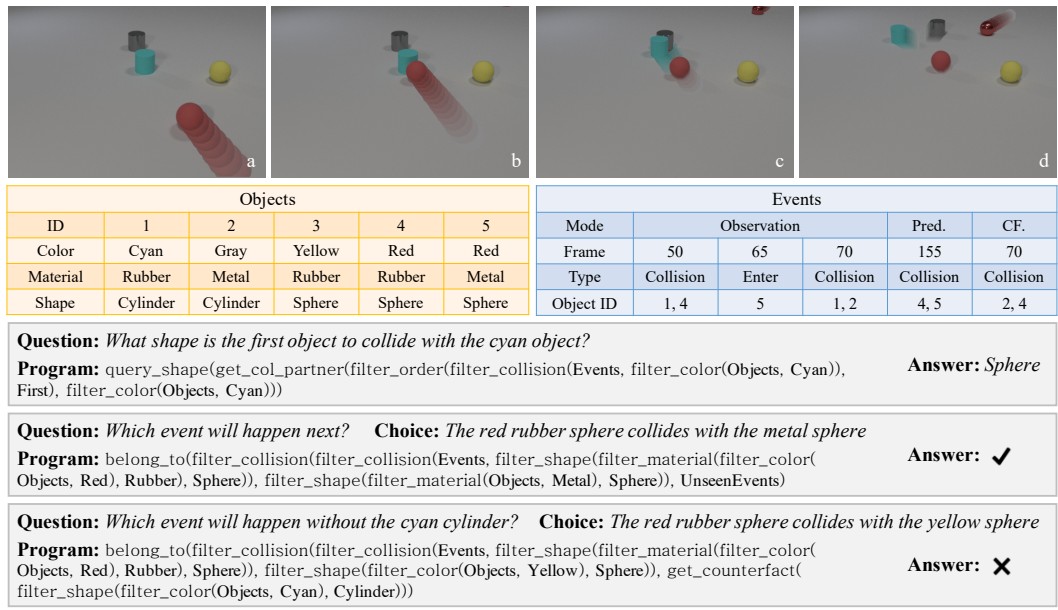

Figure 7: Sample results of NS-DR on CLEVRER. 'Pred.' and 'CF.' indicate predictive and counterfactual events extracted by the model's dynamics predictor. The counterfactual condition shown in this example is to remove the cyan cylinder.

The output of the neural dynamics predictor is $\langle\{\hat{O}_t\}_{t=1...T}, \{\hat{R}_t\}_{t=1...T}\rangle$, the collection of object states and relations across all the observed and rollout frames. From this representation, one can recover the full motion and event traces of the objects under different rollout conditions and generate a dynamic scene representation of the video. For example, if we want to know what will happen if an object is removed from the scene, we just need to erase the corresponding vertex and associated edges from the graph and rollout using the learned dynamics to obtain the traces.

Our neural dynamics predictor is trained on the *proposed* object masks and attributes from the frame parser. We filter out inconsistent object proposals by matching the object intrinsic attributes across different frames and keeping the ones that appear in more than 10 frames. For a video of length 125, we will sample 5 rollouts, where each rollout contains 25 frames that are uniformly sampled from the original video. We normalize the input data to the range of $[-1, 1]$ and concatenate them over a time window of size 3. We set the propagation step $L$ to 2, and the dimension of the propagating effects $D_{\text{enc}}$ to 512. We use the Adam optimizer (Kingma & Ba, 2015) with an initial learning rate of $10^{-4}$, and a decay factor of 0.3 per 3 epochs. The model is trained for 9 epochs with batch size 2.

**Question parser.** For open-ended descriptive questions, the question parser is trained on various numbers of randomly selected question-program pairs. For multiple choice questions, we separately train the question parser and the choice parser. Training with $n$ samples means training the question parser with $n$ questions randomly sampled from the multiple choice questions (all three types together), and then training the choice parser with $4n$ choices sampled from the same pool. All models are trained using Adam (Kingma & Ba, 2015) for 30,000 iterations with batch size 64 and learning rate $7 \times 10^{-4}$.

**Program executor.** A comprehensive list of all modules in the NS-DR program executor is summarized in Table 4 and Table 5. The input and output data types of each module are summarized in Table 6.

## C  EXTRA EXAMPLES FROM CLEVRER

We show extra examples from CLEVRER in Figure 8. We also present visualizations of question/choice programs of different question types in Figure 9 through Figure 11. The full dataset will be made available for download.

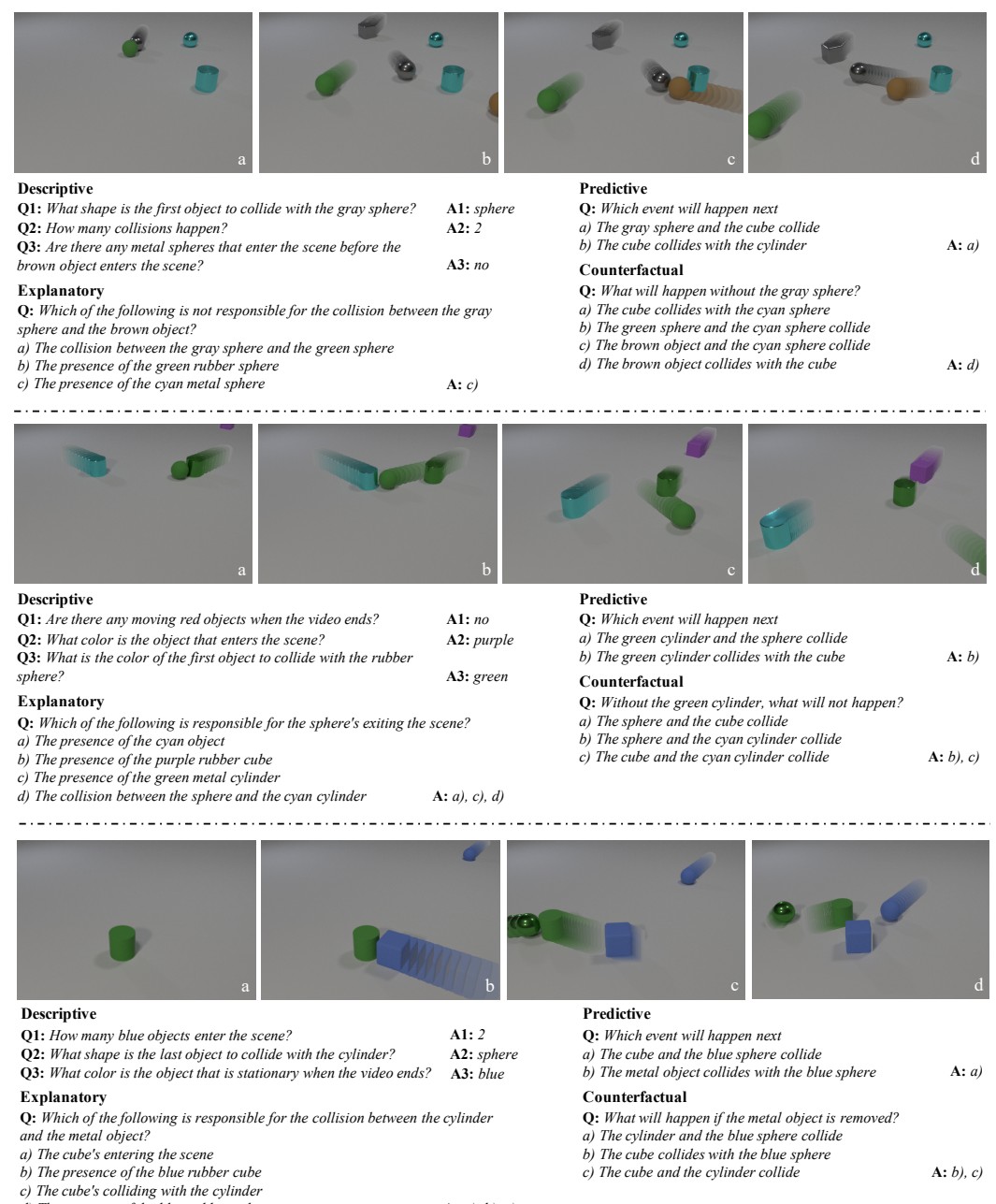

**Descriptive**
**Q1:** *What shape is the first object to collide with the gray sphere?* **A1:** *sphere*
**Q2:** *How many collisions happen?* **A2:** *2*
**Q3:** *Are there any metal spheres that enter the scene before the brown object enters the scene?* **A3:** *no*

**Explanatory**
**Q:** *Which of the following is not responsible for the collision between the gray sphere and the brown object?*
a) *The collision between the gray sphere and the green sphere*
b) *The presence of the green rubber sphere*
c) *The presence of the cyan metal sphere* **A:** *c)*

**Predictive**
**Q:** *Which event will happen next*
a) *The gray sphere and the cube collide*
b) *The cube collides with the cylinder* **A:** *a)*

**Counterfactual**
**Q:** *What will happen without the gray sphere?*
a) *The cube collides with the cyan sphere*
b) *The green sphere and the cyan sphere collide*
c) *The brown object and the cyan sphere collide*
d) *The brown object collides with the cube* **A:** *d)*

**Descriptive**
**Q1:** *Are there any moving red objects when the video ends?* **A1:** *no*
**Q2:** *What color is the object that enters the scene?* **A2:** *purple*
**Q3:** *What is the color of the first object to collide with the rubber sphere?* **A3:** *green*

**Explanatory**
**Q:** *Which of the following is responsible for the sphere's exiting the scene?*
a) *The presence of the cyan object*
b) *The presence of the purple rubber cube*
c) *The presence of the green metal cylinder*
d) *The collision between the sphere and the cyan cylinder* **A:** *a), c), d)*

**Predictive**
**Q:** *Which event will happen next*
a) *The green cylinder and the sphere collide*
b) *The green cylinder collides with the cube* **A:** *b)*

**Counterfactual**
**Q:** *Without the green cylinder, what will not happen?*
a) *The sphere and the cube collide*
b) *The sphere and the cyan cylinder collide*
c) *The cube and the cyan cylinder collide* **A:** *b), c)*

**Descriptive**
**Q1:** *How many blue objects enter the scene?* **A1:** *2*
**Q2:** *What shape is the last object to collide with the cylinder?* **A2:** *sphere*
**Q3:** *What color is the object that is stationary when the video ends?* **A3:** *blue*

**Explanatory**
**Q:** *Which of the following is responsible for the collision between the cylinder and the metal object?*
a) *The cube's entering the scene*
b) *The presence of the blue rubber cube*
c) *The cube's colliding with the cylinder*
d) *The presence of the blue rubber sphere* **A:** *a), b), c)*

**Predictive**
**Q:** *Which event will happen next*
a) *The cube and the blue sphere collide*
b) *The metal object collides with the blue sphere* **A:** *a)*

**Counterfactual**
**Q:** *What will happen if the metal object is removed?*
a) *The cylinder and the blue sphere collide*
b) *The cube collides with the blue sphere*
c) *The cube and the cylinder collide* **A:** *b), c)*

Figure 8: Sample videos and questions from CLEVRER. Stroboscopic imaging is applied for motion visualization.

| Module Type | Name / Description | Input Type | Output Type |
|---|---|---|---|
| Input Modules | `Objects`
Returns all objects in the video | - | *objects* |
| | `Events`
Returns all events that happen in the video | - | *events* |
| | `UnseenEvents`
Returns all future events after the video ends | - | *events* |
| | `AllEvents`
Returns all possible events on / between any objects | - | *events* |
| | `Start`
Returns the special "start" event | - | *event* |
| | `End`
Returns the special "end" event | - | *event* |
| Object Filter Modules | `Filter_color`
Selects objects from the input list with the input color | (*objects*, *color*) | *objects* |
| | `Filter_material`
Selects objects from the input list with the input material | (*objects*, *material*) | *objects* |
| | `Filter_shape`
Selects objects from the input list with the input shape | (*objects*, *shape*) | *objects* |
| | `Filter_moving`
Selects all moving objects in the input frame
or the entire video (when input frame is "null") | (*objects*, *frame*) | *objects* |
| | `Filter_stationary`
Selects all stationary objects in the input frame
or the entire video (when input frame is "null") | (*objects*, *frame*) | *objects* |
| Event Filter Modules | `Filter_in`
Selects all incoming events of the input objects | (*events*, *objects*) | *events* |
| | `Filter_out`
Selects all exiting events of the input objects | (*events*, *objects*) | *events* |
| | `Filter_collision`
Selects all collisions that involve any of the input objects | (*events*, *objects*) | *events* |
| | `Filter_before`
Selects all events before the target event | (*events*, *event*) | *events* |
| | `Filter_after`
Selects all events after the target event | (*events*, *event*) | *events* |
| | `Filter_order`
Selects the event at the specific time order | (*events*, *order*) | *event* |
| | `Filter_ancestor`
Selects all causal ancestors of the input event | (*events*, *event*) | *events* |
| | `Get_frame`
Returns the frame of the input event in the video | *event* | *frame* |
| | `Get_counterfact`
Selects all events that would happen if the object is removed | (*events*, *object*) | *events* |
| | `Get_col_partner`
Returns the collision partner of the input object
(the input event must be a collision) | (*event*, *object*) | *object* |
| | `Get_object`
Returns the object that participates in the input event
(the input event must be a incoming / outgoing event) | *event* | *object* |
| | `Unique`
Returns the only event / object in the input list | *events /*
*objects* | *event /*
*object* |

Table 4: Functional modules of NS-DR's program executor.

| Module Type | Name / Description | Input Type | Output Type |
|---|---|---|---|
| Output Modules | `Query_color`
Returns the color of the input object | *object* | *color* |
| | `Query_material`
Returns the material of the input object | *object* | *material* |
| | `Query_shape`
Returns the shape of the input object | *object* | *shape* |
| | `Count`
Returns the number of the input objects | *objects* | *int* |
| | `Exist`
Returns "yes" if the input objects is not empty | *objects* | *bool* |
| | `Belong_to`
Returns "yes" if the input event belongs to the input set of events | (*event, events*) | *bool* |
| | `Negate`
Returns the negation of the input boolean | *bool* | *bool* |

Table 5: Functional modules of NS-DR's program executor (continued).

| Data type | Description |
|---|---|
| *object* | A dictionary storing the intrinsic attributes of an object in a video |
| *objects* | A list of *object*s |
| *event* | A dictionary storing the type, frame, and participating objects of an event |
| *events* | A list of *event*s |
| *order* | A string indicating the chronological order of an event out of "first", "second", "last" |
| *color* | A string indicating a color out of "gray", "brown", "green", "red", "blue", "purple", "yellow", "cyan" |
| *material* | A string indicating a material out of "metal", "rubber" |
| *shape* | A string indicating a shape out of "cube", "cylinder", "sphere" |
| *frame* | An integer representing the frame number of an event |

Table 6: Input/output data types of modules in the program executor.

```
Question: "How many spheres are moving when the
video ends?"
Count(
    Filter_move(
        Filter_shape(Objects, "Sphere"),
        Get_frame(End)
    )
)

Question: "Are there any stationary spheres when the
metal sphere enters the scene?"
Exist(
    Filter_stationary(
        Filter_shape(Objects, "Sphere"),
        Query_frame(
            Filter_in(
                Events,
                Filter_shape(
                    Filter_material(Objects, "Metal"),
                    "Sphere"
                )
            )
        )
    )
)
```

```
Question: "What shape is the second object to collide with
the green object?"
Query_shape(
    Get_col_partner(
        Filter_order(
            Filter_collision(
                Events,
                Filter_color(Objects, "Green")
            ),
            "Second"
        ),
        Filter_color(Objects, "Green")
    )
)

Question: "What color is the object to collide with the blue
object?"
Query_color(
    Get_col_partner(
        Filter_collision(
            Events,
            Filter_color(Objects, "Blue")
        ),
        Filter_color(Objects, "Blue")
    )
)
```

Figure 9: Example of descriptive question programs.

Question: "Which of the following is not responsible for the metal sphere's colliding with the rubber sphere?"

```
Negate(
   Belong_to(
      Event, // Output of the Choice program
      Filter_ancestor(
         Filter_collision(
            Filter_collision(
               Events,
               Filter_shape(
                  Filter_material(Objects, "Metal"),
                  "Sphere"
               )
            ),
            Filter_shape(
               Filter_material(Objects, "Rubber"),
               "Sphere"
            )
         )
      )
   )
)
```

Choice: "The collision between the green cube and the rubber sphere"

```
Filter_collision(
   Filter_collision(
      Events,
      Filter_shape(
         Filter_color(Objects, "Cube"),
         "Cube"
      )
   ),
   Filter_shape(
      Filter_material(Objects, "Rubber"),
      "Sphere"
   )
)
```

Choice: "The green cube's entering the scene"

```
Filter_in(
   Filter_shape(
      Filter_color(Objects, "Green"),
      "Cube"
   )
)
```

Figure 10: Example of explanatory question and choice programs.

Counterfactual question: "Which event will happen if the rubber sphere is removed?"

```
Belong_to(
   Event, // Output of the choice program
   Get_counterfact(
      Filter_shape(
         Filter_material(Objects, "Rubber"),
         "Sphere"
      )
   )
)
```

Predictive question: "What will happen next?"

```
Belong_to(
   Event, // Output of the choice program
   UnseenEvents
)
```

Choice: "The cube collides with the metal sphere"

```
Filter_collision(
   Filter_collision(
      AllEvents,
      Filter_shape(Objects, "Cube")
   ),
   Filter_shape(
      Filter_material(Objects, "Metal"),
      "Sphere"
   )
)
```

Choice: "The metal sphere and the cylinder collide"

```
Filter_collision(
   Filter_collision(
      AllEvents,
      Filter_shape(
         Filter_material(Objects, "Metal"),
         "Sphere"
      )
   ),
   Filter_shape(Objects, "Cylinder")
)
```

Figure 11: Example of predictive / counterfactual question and choice programs.

