# OpenReview forum: "CLEVRER: Collision Events for Video Representation and Reasoning"
_ICLR.cc/2020/Conference — Accept (Spotlight)_

### Official Review · AnonReviewer1 · 2019-10-19
**Official Blind Review #1**

**Rating:** 6

**Review:**

A new benchmark is presented, which requires to reason on spatio-temporal data (videos) and incorporates physics (videos of physical dynamics), language (questions are formulated through language) and causality. The benchmark builds on the well-known CLEVR benchmark and adds several interesting contributions, in particular reasoning over time.

Compared to other benchmarks proposed on reasoning on physical dynamics, this benchmark adds to the language component (which is presented in classical VQA benchmarks), and an interesting counterfactual component known from the causal inference literature. This is fairly new for the computer vision and statistical machine learning literature (a few papers on counterfactual reasoning exist), but in contrast to the causal inference literature, here the do-operator is observable during training: we do have access to do-interventions and even the outcome during training and therefore we can even learn counterfactuals using supervised learning. This statement is not meant as criticism, as learning counterfactuals without supervision from high-dimensional input like images seems to be currently out of a reach, or at least has not yet been demonstrated up to my knowledge.

On the downside, compared to other physical reasoning benchmarks, the answers here are multiple choice instead of regressions of the future motion, which requires more fine-grained reasoning. This will guide solutions to a certain type and will also favor solutions, requiring the network to detect certain binary concepts and combine them instead of regressing complex functions. It also favors solutions of the type presented in the paper.

As for other benchmarks, the dataset is quite large (20 000 videos) and, given its synthetic nature, is accompanied by functional programs. As for CLEVR, the simulations have been performed by a physics engine and then separately rendered with Blender to maximize visual quality. The result is a very interesting benchmark, and I have no doubt that it will be very useful for us (the learning reasoning community).

Another positive point is the number of baselines tested on the benchmark, among which we can find strong papers on VQA/VQA2

I have a couple of questions:

How is the causal graph created? Are the experiments rendered and outcomes examined, creating the causal graph for the counterfactual answers, or are the experiments selected with a given outcome already decided?

The distribution of question types has been provided, but how about the biases? Distributions of the answers would have been helpful. How did the authors avoid biases during construction of the dataset, in particular in the counterfactual case (see remarks on the causal graph).

The paper also comes with a method for solving, which is very similar to existing methods on learning through functional programs. The method itself is unfortunately described only very briefly and the reader needs to look it up almost entirely in the appendix of the paper, which is surprising, as the paper is only 8 pages long instead of 10.

As for CLEVR, one of the downsides of this type of benchmark is the synthetic nature of the images and the limited range of different objects in the scene. Of course this comes with the advantage of being able to study compositional reasoning in detail, as a scene graph can be calculated easily (and is available during simulation). However, it also makes reasoning through functional programs much easier, as the proposed filters are limited in number and can strong respond to the small number of shapes and colors available in the data. I have strong doubts that this kind of approach extends to real life scenarios.

For VQA type of scenarios, GQA is a nice compromise between natural looking images and the availability of scene graphs and the restriction of questions to compositional reasoning. The optimal choice would be a similar compromise for spatio-temporal data, but of course this would be a huge effort and it would be up to impossible to have access to counterfactuals.

Last point, and this question is not restricted to this paper, as the name came up elsewhere, why is the model called neuro-symbolic reasoning? While it could be argued that the questions require a sort of “symbolic reasoning”, I am not sure that the reasoning method itself is symbolic even partially. Other than selecting functional programs out of a discrete set, the reasoning itself is connectionist and performed with graph networks.


**Experience Assessment:**

I have published one or two papers in this area.

**Review Assessment: Checking Correctness Of Derivations And Theory:**

I carefully checked the derivations and theory.

**Review Assessment: Checking Correctness Of Experiments:**

I carefully checked the experiments.

**Review Assessment: Thoroughness In Paper Reading:**

I read the paper thoroughly.

---

> ### Author Response · Authors · 2019-11-13
> **Response to Reviewer #1**
>
> Thanks a lot for your helpful comments and suggestions about our manuscript. We address your specific concerns and questions below.
>
> 1. Multiple choice versus fine-grained physical reasoning.
>
> Even though the tasks in CLEVRER are grounded to multiple choices on natural language inputs, identification and prediction of collision events require precise prediction of the motion trajectories, since otherwise the moving objects will miss each other. In this sense, fine-grained physical reasoning is an implicit requirement of our task. This is also shown by the fact that having a dynamics model for trajectory prediction leads to stronger results than other baselines.
>
> Also, CLEVRER provides detailed annotations of the object motion trajectories for diagnostics and further benchmarking, based on which new evaluation metrics such as trajectory accuracy can be easily added to the dataset.
>
> 2. Generation of the causal events.
>
> The causal events in the dataset are created by a recursive process as stated in section 3.1. To further specify the details, we start with simulating the motion of one single object, and then add another object whose initial position / velocity is set such that it will lead to a collision between the two objects. We repeat this process to add more objects to collide with either or both of the outgoing objects from previous collisions. In this way, a causal relation between the collision events is established. The videos are rendered from the final simulation outputs with annotated causal structure.
>
> 3. Answer distribution and question biases.
>
> The answer distribution of descriptive questions can be found in supplementary material A (figure 6), which indicates a very well-balanced distribution. The number of correct choices versus incorrect choices in the multiple-choice questions is 1:1. As shown in Table 2, the random baseline has ~50% per-option accuracy, which suggests the number of correct and incorrect choices are balanced.
>
> We carefully designed the question generation process to minimize biases at various levels. As also mentioned in the general response #1, for each video and question type, we sample the questions to balance the answer space. Furthermore, resampling the questions also reduces the language bias, as the answers might be statistically correlated with the length of the questions or co-occur with certain word combinations. We apply rejection sampling to maintain an uncorrelated joint distribution between the questions and answers over the entire dataset. The result of the LSTM model shows a small performance gain over the random baselines, which suggest the language bias is effectively removed. Finally, for the causal events, the motion trajectories and collision targets are fully randomized and independent to the visual attributes (color, shape, material) during the generation process of the causal graph as described above. As a result, undesired correlations are prevented and the questions have small bias.
>
> 4. Model details and paper length.
>
> Thanks, we will follow your suggestion to move more key details of the model from the supplementary materials to the main text.
>
> 5. Extension to real-life scenarios.
>
> As also mentioned in the general response #2, our approach of incorporating an object-centric dynamics model for physical reasoning has similar applications in robotics planning and manipulation [Janner et al. 2018] [Veerapaneni et al. 2019]. We also agree that combining the design principles of CLEVRER with GQA [Hudson and Manning 2019] is a great way of further extending our task to complex real-world scenarios. We hope to note that the tasks in CLEVRER still remain challenging to current approaches, and a stronger model on the dataset will contribute towards causal reasoning in more complex real environments.
>
> 6. Further clarifications on the term “neuro-symbolic”
>
> The word “symbolic” in our framework corresponds to symbolic program execution, during which the inferred question logic (program) is explicitly applied to the extracted motion and event traces. This process is operated on symbolic representations. In a more general sense, “neuro-symbolic” stands for using neural network for pattern recognition and symbolic operations for logic reasoning, which combines the best of two approaches.
>
> - Janner, Michael, et al. "Reasoning about physical interactions with object-oriented prediction and planning." ICLR (2019).
> - Veerapaneni, Rishi, et al. "Entity Abstraction in Visual Model-Based Reinforcement Learning." CoRL (2019).
> - Hudson, Drew A., and Christopher D. Manning. "GQA: A new dataset for real-world visual reasoning and compositional question answering." CVPR (2019).

---

### Official Review · AnonReviewer2 · 2019-10-22
**Official Blind Review #2**

**Rating:** 8

**Review:**

This paper studies the temporal and causal structures in videos. Specifically, the authors first introduce a new dataset called CLEVRER drawing motivation from CLEVR, a well-known visual reasoning dataset. They further evaluate a set of state-of-the-art methods on the newly introduced dataset to confirm their initial beliefs of the challenges posed by causal reasoning. Based on empirical clues, they also suggest neural-symbolic based framework for causal reasoning.
On the whole, I think this is a good paper and it addresses one of the most challenging and exciting tasks in visual reasoning. The introduction of the CLEVRER dataset is likely valuable for the community to facilitate research in this area. I have some concerns as follows:
(1) As the fact that questions are generated algorithmically, I believe many of them are either ill-posed or degenerate similar to what described in the CLEVR. Did you manage to filter out those questions? Please provide more details on the question generation process.
(2) I have a doubt on the reported results in table 2 as you extract visual features not fairly between all methods. Outputs of pool5 feature basically kill all spatial information while 14x14 feature map used as input of MAC, IEP and TbD-net keep the spatial information. I also not sure if the temporal attention is a better option than vectorizing the whole video features (Some pooling layers might be helpful) before feeding into those methods. It would be fairer to evaluate the state of the art methods on the CLEVRER more carefully.
(3) As the number of objects in CLEVRER is limited, using class labels from Mask R-CNN may introduce noises to the model due to incorrect detections. The authors may need to explain more clearly at this point in the implementation details.
(4) Have you tried to incorporate flow features, for example, C3D/I3D feature?

Minor comments:
There are some typos in the paper in both main paper and supplementary document -causal vs. casual. Please fix these.

**Experience Assessment:**

I have published one or two papers in this area.

**Review Assessment: Checking Correctness Of Derivations And Theory:**

I did not assess the derivations or theory.

**Review Assessment: Checking Correctness Of Experiments:**

I assessed the sensibility of the experiments.

**Review Assessment: Thoroughness In Paper Reading:**

I read the paper at least twice and used my best judgement in assessing the paper.

---

> ### Author Response · Authors · 2019-11-13
> **Response to Reviewer #2**
>
> Thanks a lot for your helpful comments and suggestions about our manuscript. We address your specific concerns and questions below.
>
> 1. Overcoming ill-posed and degenerate questions
>
> We provide further details of the question generation process in our general response #1. The ill-posed and degenerate questions in CLEVR is caused by the high order logic applied to the object attributes (i.e. “What is the color of the sphere left to the cylinder right to the green sphere?”). However in CLEVRER, the logic traces of the questions span over multiple domains including object attributes, temporal order, event attributes, etc (i.e. “What is the color of the sphere that first collides with the cylinder?”). We intentionally avoid applying high-order logic within the same domain. This effectively removes the degeneracy and prevents ill-posed questions.
>
> 2. Features for baseline evaluation.
>
> Thanks a lot for the suggestion on improving the baselines. Following your suggestion, we conducted experiments on the CNN-based methods by replacing the pool5 feature by both the 14 x 14 and the I3D feature maps. We also replaced LSTM with a ConvLSTM to preserve spatial information. The results are summarized below:
>
> - CNN (14 x 14) + ConvLSTM:
> Descriptive: 58%
> Explanatory (per-opt / per-Q): 63.4% / 16.8%
> Predictive (per-opt / per-Q): 56.3% / 29.8%
> Counterfactual (per-opt / per-Q): 63.1% / 13.6%
>
> - I3D + ConvLSTM:
> Descriptive: 62%
> Explanatory (per-opt / per-Q): 62.2% / 23.3%
> Predictive (per-opt / per-Q): 51.8% / 36.4%
> Counterfactual (per-opt / per-Q): 58.5% / 11.3%
>
> These methods still do not show strong performance on the causal reasoning tasks (explanatory, predictive, and counterfactual), which is consistent to our observation that incorporating a dynamics model is essential to the task.
>
> 3. Model noise from Mask R-CNN misdetections.
>
> Our video frame parser is inspired by the scene parser from [Yi et al. 2018], which achieves strong performance on the CLEVR dataset. We use their open-sourced code (https://github.com/kexinyi/ns-vqa) for scene parsing on the same visual domain (CLEVRER uses the same renderer as CLEVR). In practice, Mask-RCNN performs well and misdetections are rare. On the other hand, since our neuro-symbolic framework is disentangled and operates on grounded representations, the errors caused by the misdetections are transparent and can be diagnosed.
>
> References:
> - Yi, Kexin, et al. "Neural-Symbolic VQA: Disentangling reasoning from vision and language understanding." NeurIPS (2018).

---

### Official Review · AnonReviewer3 · 2019-10-25
**Official Blind Review #3**

**Rating:** 6

**Review:**

Authors propose a new Dataset CLEVRER, a simulated video dataset involving interaction between objects. It is discussed, the existing state-of-the-art models for visual question answering, doesn’t capture the causal structure between the objects and their claim is supported by their experiments.  Authors also proposed a model which captures the dynamics of the objects involved in the video, through experiments they have shown their model performs better than the existing models.

The CLEVRER dataset is designed to test a model capability to answer the queries involving causal relationship between the objects involved in the video.

Details.
This work begins with a well motivated problem by pointing out the drawback in existing VQA(Visual Question Answering)  models, that existing works focus on visual and input language patterns to answer the queries and doesn’t tackle the task involving causal structure.  It  explores the current literature around the problem related to visual question answering(both real world data and simulated data). Through experiments they have shown the existing state-of-the-art work on visual question answering doesn’t perform well on the dataset CLEVRER.

To prove the incompetence of existing models to capture causal structure, authors designed an artificial dataset with questions which can be answered only when the model is capable of capturing the causal structure between the objects.

The process involving the creation of CLEVRER dataset is well explained. But, it is unclear how the questions are generated.

Through experiments authors revealed the drawbacks of existing models on capturing the dynamics between the objects and  proposed a model which is said to be inspired by previous VQA[1] model. An important modification by incorporating neural dynamics predictor module to the existing model is key, and also achieves good performance on the dataset.

Comments:

- The paper is well written, but it is unclear how the questions are generated during dataset creation process.
- The main contribution of the paper is to show the incompetence of existing models to capture dynamics. Which is a form of analysis.
- It is shown that,  learning dynamics of the objects the model can achieve better performance.
- This dataset is created in more restricted environment like height of the objects should be same. How can this be generalized to a more real world setting ?

Some minor issues:
In few places causal is misspelled as casual(page 2,8).
Equation 2, in the appendix the subscripts are not proper.

[1] Yi, Kexin, et al. "Neural-symbolic vqa: Disentangling reasoning from vision and language understanding." Advances in Neural Information Processing Systems. 2018.

**Experience Assessment:**

I do not know much about this area.

**Review Assessment: Checking Correctness Of Derivations And Theory:**

N/A

**Review Assessment: Checking Correctness Of Experiments:**

I assessed the sensibility of the experiments.

**Review Assessment: Thoroughness In Paper Reading:**

I made a quick assessment of this paper.

---

> ### Author Response · Authors · 2019-11-13
> **Response to Reviewer #3**
>
> Thanks a lot for your helpful comments and suggestions about our manuscript. We address your specific concerns and questions below.
>
> 1. Details of question generation.
>
> As also mentioned in the general response #1, questions in CLEVRER are generated by a multi-step procedure. For each question type, a pre-defined logic template is chosen. The logic template can be further populated by attributes that are specifically associated with the video context (i.e. the color, material, shape that identifies the object to be queried). For a given video and template, we first generate the set of all possible questions by exhaustively iterate through all possible attributes. Then we sample from the list of questions to balance answer distribution and minimize language biases.
>
> 2. Generalize to real world setting.
>
> As also discussed in the general response #2, our approach of incorporating an object-centric dynamics model for physical reasoning has similar applications in robotics planning and manipulation [Janner et al. 2018] [Veerapaneni et al. 2019]. Furthermore, incorporating temporal and causal annotations to real videos similar to the GQA dataset [Hudson and Manning 2019] is also an important direction to pursue. We also hope to note that, even within the restricted domain, the causal reasoning tasks still remain challenging for current approaches. Further studies on these tasks under a controlled environment will contribute towards building a model for real-world causal reasoning.
>
> References:
> - Janner, Michael, et al. "Reasoning about physical interactions with object-oriented prediction and planning." ICLR (2019).
> - Veerapaneni, Rishi, et al. "Entity Abstraction in Visual Model-Based Reinforcement Learning." CoRL (2019).
> - Hudson, Drew A., and Christopher D. Manning. "GQA: A new dataset for real-world visual reasoning and compositional question answering." CVPR (2019).

---

### Author Response · Authors · 2019-11-13
**General Response**

We thank all reviewers for their constructive comments and suggestions to improve the strength and clarity of this paper. Here we address some common concerns raised by the reviewers. We will update the text of the paper to further incorporate these suggestions and fix the typos.

1. Details of question generation (R1, R2, R3)

Questions in CLEVRER are generated by a multi-step procedure. For each question type, a pre-defined logic template is chosen. The logic template can be further populated by attributes that are specifically associated with the video context (i.e. the color, material, shape that identifies the object to be queried). For a given video and template, we first generate the set of all possible questions by exhaustively iterate through all possible attributes. Then we sample from the list of questions to balance answer distribution and minimize language biases.

2. Relation to real-world tasks and applications (R1, R3)

Our approach of incorporating an object-centric dynamics model in physical reasoning is related to applications in robotic planning and manipulation. For example, [Janner et al. 2018] and [Veerapaneni et al. 2019] have demonstrated using object-centric dynamics model for planning in real-world manipulation tasks.

There are still several steps to take towards causal reasoning in complex and noisy real environments. Along this path, CLEVRER is designed to serve as an initial step that focuses on the temporal and causal relations under a constrained visual context. The environment also supports adding richer and more realistic visual and physical entities, such as object texture, size, elasticity, friction, etc. As suggested by Reviewer 1, a video dataset similar to GQA [Hudson and Manning 2019] would be a very nice goal to aim for as the next step.

We also emphasize that the tasks in CLEVRER remain challenging in its current form. As shown in Table 2, current deep learning models do not perform well on CLEVRER, especially for counterfactual tasks: the best baseline (without a dynamics model) achieves a per-option accuracy of 63.5%, while NS-DR (with a dynamics model) achieves 74.1%. A stronger model on CLEVRER will contribute toward causal reasoning in more complex real environments.

References:
- Janner, Michael, et al. "Reasoning about physical interactions with object-oriented prediction and planning." ICLR (2019).
- Veerapaneni, Rishi, et al. "Entity Abstraction in Visual Model-Based Reinforcement Learning." CoRL (2019).
- Hudson, Drew A., and Christopher D. Manning. "GQA: A new dataset for real-world visual reasoning and compositional question answering." CVPR (2019).

---

### Decision · Program_Chairs · 2019-12-19

**Decision:**

Accept (Spotlight)

**Comment:**

The reviewers are unanimous in their opinion that this paper offers a novel approach to causal learning.  I concur.